# On the engineering of higher-order Van Hove singularities in two dimensions

Anirudh Chandrasekaran [1] ✉, Luke C. Rhodes [2], Edgar Abarca Morales[2,3], Carolina A. Marques [2,4], Phil D. C. King [2], Peter Wahl [2,5] & Joseph J. Betouras[1] ✉

The properties of correlated electron materials are often intricately linked to Van Hove singularities (VHS) in the vicinity of the Fermi energy. The class of these VHS is of great importance, with higher-order ones—with power-law divergence in the density of states—leaving frequently distinct signatures in physical properties. We use a new theoretical method to detect and analyse higher-order VHS (HOVHS) in two-dimensional materials and apply it to the electronic structure of the surface layer of $Sr_2RuO_4$. We then constrain a low energy model of the VHS of the surface layer of $Sr_2RuO_4$ against angle-resolved photoemission spectroscopy and quasiparticle interference data to analyse the VHS near the Fermi level. We show how these VHS can be engineered into HOVHS.

Technological breakthrough of modern electronics hinges on our ability to gain control over exotic properties of materials with strong electron correlations. The control, however, has proven to be very difficult without a qualitative and quantitative theoretical picture behind the observed behaviour. The properties of strongly correlated electron materials can be extremely sensitive to modest external stimuli, such as magnetic field, pressure or uniaxial strain, often exhibiting complex phase diagrams as a function of these tuning parameters[1-4]. In many cases, the sensitivity of their properties is traced back to van Hove singularities (VHS) in the vicinity of the Fermi energy ($E_F$)[5-8]. It has been realised that such VHS can also be of a higher-order type (HOVHS), where both the gradient and the Hessian determinant of the dispersion relation $\varepsilon(k_x, k_y)$ vanish with unique thermodynamic signatures as well as unconventional phase formation[5]. HOVHS and the corresponding flattening of the single-particle band structure are important due to the crucial role that interactions play as the Fermi velocity tends to zero. More importantly, HOVHS have been now observed in a range of materials in addition to ruthenates[5] and twisted bilayer graphene[7] such as kagome metals and superconductors[9,10], doped graphene[11] and may be relevant to the observed phases of the Bernal type bilayer graphene[12]. The ability to design materials with desired properties is heavily reliant on the deep understanding of all essential aspects of the physics. In the present case, we focus on HOVHS and take as a paradigm the surface of the well-studied material $Sr_2RuO_4$.

In explaining the thermodynamic properties of $Sr_3Ru_2O_7$[5], the concept of a multicritical Fermi surface topological transition was used as a new qualitative concept. DFT calculations backed the assumption of a specific type of HOVHS ($X_9$) without further analysis due to lack of precise theoretical tools. In the meantime, a detailed classification scheme of HOVHS was developed[13] and the effects of disorder were studied[14].

Recently, a new method to detect and analyse a HOVHS from arbitrary two dimensional electronic structure models has been developed[15], enabling us to uniquely characterise the important band structure features in two dimensional systems. The method expands the dispersion relation directly in the vicinity of the VHS, and utilises the Hellman-Feynman theorem to evaluate higher-order derivatives. Extending the method to real materials requires a detailed understanding of the material's low-energy electronic structure. The key difference in the present work is that we use a fully ab-initio approach to model the electronic structure and apply an assumption free method to determine the order of the HOVHS that is applicable, in principle, to any material. The surface of $Sr_2RuO_4$ provides an ideal test

---

[1]Department of Physics and Centre for the Science of Materials, Loughborough University, Loughborough, UK. [2]SUPA, School of Physics and Astronomy, University of St Andrews, North Haugh, St Andrews, UK. [3]Max Planck Institute for Chemical Physics of Solids, Dresden, Germany. [4]Physik-Institut, Universität Zürich, Zürich, Switzerland. [5]Physikalisches Institut, Universität Bonn, Bonn, Germany. ✉e-mail: A.Chandrasekaran@lboro.ac.uk; J.Betouras@lboro.ac.uk

system to explore how the symmetry and order of the VHS influences the electronic properties of multi-band and spin−orbit coupled (SOC) systems. Indeed, strontium ruthenates provide an ideal benchmark material class to explore the impact of VHS on macroscopic properties[1]. They host rich phase diagrams, with metamagnetic phases, a putative quantum critical point in $Sr_3Ru_2O_7$[16] and evidence for unconventional superconductivity in $Sr_2RuO_4$[17]. In both systems, there is strong evidence for the importance of VHS in their electronic properties[8,18−22]. The superconductivity in $Sr_2RuO_4$, for example, has been found to react sensitively to uniaxial strain[4] which directly correlates with a VHS crossing $E_F$[8], while in $Sr_3Ru_2O_7$, an unusual field dependence of the specific heat has been explained from Zeeman splitting of HOVHS[5].

The surface layer of $Sr_2RuO_4$ provides a two-dimensional electronic system on which high-resolution spectroscopic data is available[20,23−25], facilitating a full assessment of the nature of its VHS. A $\sqrt{2} \times \sqrt{2}$ structural reconstruction due to octahedral rotations[20,23,26,27] results in a unit cell doubling that produces a four-fold saddle point in the electronic structure at the Brillouin zone (BZ) corner. This type of saddle point is an essential ingredient for the formation of a HOVHS (of classification type $X_9$[13]) in $Sr_3Ru_2O_7$[5] with power-law divergence in the density of states (DoS), distinct from the logarithmic divergence of the $A_1$ saddle point at the BZ face in the bulk of $Sr_2RuO_4$.

In this work, we establish analytical and numerical tools to study the order of VHSs in realistic band structures as obtained from DFT calculations and apply these, as a demonstration of the capabilities of these methods, to how the octahedral rotation influences the formation of HOVHS in the surface layer of $Sr_2RuO_4$. Furthermore, we show that different perturbations may lead to different type of HOVHS which facilitates the engineering of HOVHS. We first study the evolution of the VHS from density functional theory (DFT) calculations and establish the impact of octahedral rotations on the VHS, before constraining a minimal model directly against experimental data from angle-resolved photoemission spectroscopy (ARPES) and scanning tunnelling microscopy (STM). By combining these complementary data sets, we are able to progressively refine the low-energy electronic structure and obtain an accurate description of the dispersion relations in the vicinity of the saddle point. From the experimentally-derived model, we classify multiple VHS close to $E_F$ and study how their order can be tuned, providing new design principles for VHS engineering in two-dimensional materials.

## Results

### Octahedral rotation-induced order of VHS

To analyse the impact of octahedral rotation on the electronic structure of $Sr_2RuO_4$, we perform DFT calculations of a monolayer of $Sr_2RuO_4$, shown in Fig. 1a, with induced rotation of the $RuO_6$ octahedra, keeping the Ru-Ru distances constant. This produces a two-atom unit cell (Fig. 1a). We then calculate the electronic structure for each rotation angle and project onto a tight-binding model[28]. There are two dominant changes as a function of octahedral rotation. First is the evolution of the saddle point VHS that lies close to $E_F$ at the $\overline{M}$ point. This band undergoes a Lifshitz transition at an octahedral rotation angle of ~ 5° and evolves from a concave dispersion to a convex dispersion as rotation angle increases. The second feature is the evolution and Lifshitz transition of an electron pocket at the $\overline{\Gamma}$ point. The energy of this band minimum is, however, known to be located too low in energy in DFT, with ARPES measurements placing it just above $E_F$[23,27]. This only produces a linear offset to the DoS of the VHS around $\overline{M}$ without affecting the order and symmetry of the relevant VHS around $\overline{M}$ points. Here we focus only on the electronic structure of the low-energy VHS around the $\overline{M}$ points. It is known that inclusion of electron correlations via the DFT + U scheme shifts the pocket at the $\overline{\Gamma}$ point above the Fermi energy[29].

Figures 1d and 2a, b show the electronic structure close to the $\overline{M}$ point and throughout the surface BZ, respectively, for a 9° octahedral rotation. The octahedral rotation significantly changes the curvature of the electronic structure from convex to concave in the vicinity of $\overline{M}$ point. This continuous tuning of the band dispersion leads to favourable conditions for the formation of HOVHS. By applying our method[15] to extract the order and symmetry of the VHS around the $\overline{M}$ point, we find that close to $\theta \sim 9°$, quadratic terms in the dispersion relation are suppressed in the $\overline{\Gamma} - \overline{M}$ direction and the analytic expansion predicts a HOVHS with an energy dependence in the tail of the DoS $\rho(E) \propto E^{-1/4}$, classified as $A_3$-type[13,30]. For octahedral rotation angles below/above 9°, the dispersion is dominated by quadratic terms resulting in a VHS with

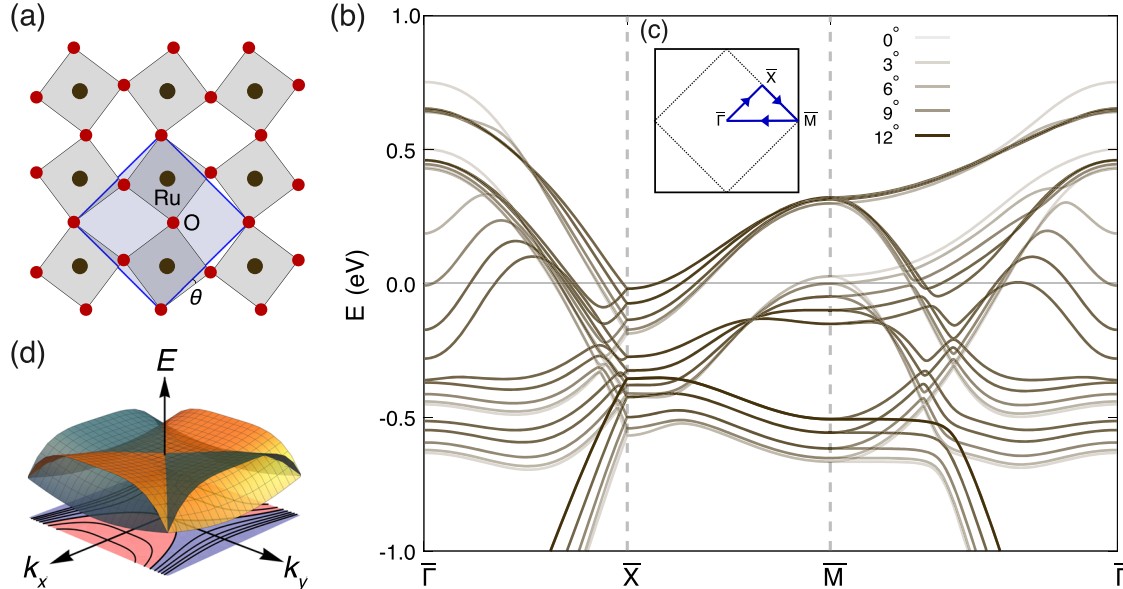

**Fig. 1 | VHS in ruthenates. a** top-down view of a single layer of $Sr_2RuO_4$ highlighting the surface-induced octahedral rotation $\theta$. Grey square shows the unit cell without octahedral rotations, the blue shaded square highlights the expanded unit cell with rotations. **b** Evolution of the band structure as a function of $\theta$. Darker lines correspond to larger Ru-O-Ru rotation angles. **c** BZ of $Sr_2RuO_4$ with octahedral rotations (dashed) and without (solid). **d** Fermi contours and 3D image of an $A_3$ HOVHS (and a $\pi/2$ rotated copy required by symmetry) that can occur at the M-point due to the reconstructed BZ.

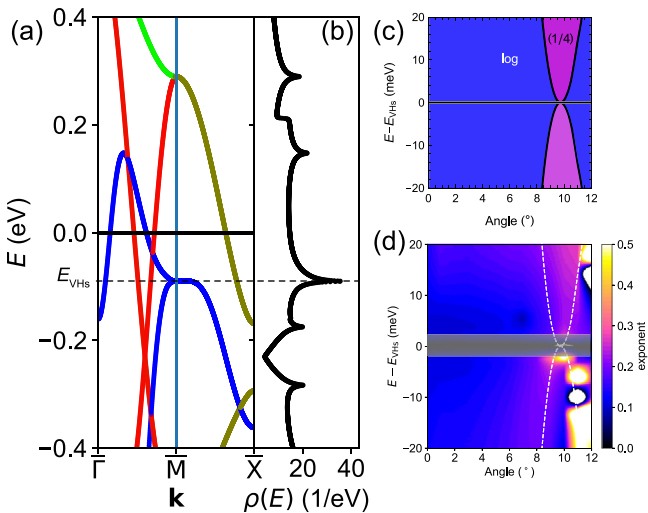

**Fig. 2 | Orbital character of the bands and the crossover from log to power-law in the density of states. a** Band structure in the vicinity of the M point showing the orbital character of VHS at a rotation angle $\theta = 9°$. Colours encode orbital character (red: $d_{xz}$, green: $d_{yz}$, blue: $d_{xy}$). The energy $E_{VHS}$ of the $d_{xy}$ VHS is indicated by a horizontal dashed line. **b** DoS for the band structure shown in (**a**), with the VHS on the $d_{xy}$ band. **c** Type of the $d_{xy}$ VHS determined from the series expansion[13] as function of angle and energy from the VHS, $E - E_{VHS}$. At an angle of $\theta = 9°$, the behaviour exhibits a singular point at $E_{VHS}$ where the VHS shows power-law divergence with exponent $-\frac{1}{4}$. **d** Numerically determined order of the tail of the VHS from $\frac{d\log\rho}{d\log E}$ for the $d_{xy}$-derived VHS as function of angle and energy. The energy scale is relative to the energy $E_{VHS}$ at which the VHS occurs. The numerically determined order is not reliable in the grey area, because it is too close to the VHS. The dashed line shows the superimposed boundary between logarithmic and polynomial order of the tail of the VHS determined analytically.

a logarithmic divergence. Fig. 2c shows this phase diagram, as a function of energy and octahedral rotation angle.

The result demonstrates that modification of the octahedral rotation provides a method to engineer HOVHS in $Sr_2RuO_4$. However, in multi-band systems there is a possibility that other electronic states could mask the clean divergent signatures induced by HOVHS in thermodynamic properties. To check this possibility and establish the energy range over which the HOVHS impacts experimental observables, we numerically integrate the DoS $\rho(E)$ calculated using the tight-binding model. The exponent of the VHS can then be determined from the logarithmic derivative of $\rho(E)$, $\frac{d\log\rho(E-E_{VHS})}{d\log E}$, which provides the order of the leading polynomial term in the tail of the VHS. The numerical analysis, shown in Fig. 2d, confirms that the DoS tail from the VHS changes significantly as a function of rotation angle. While for small octahedral rotation angles, we observe a behaviour consistent with a logarithmic divergence, as the angle $\theta$ approaches $9°$, the VHS acquires an exponent $-1/4$, consistent with the analytic theory.

## Fitting to ARPES

Having identified that HOVHS can be stabilised via octahedral rotations in monolayer $Sr_2RuO_4$, we turn our attention to a system that exhibits a series of VHS close to $E_F$ with the potential to tune them, controlling their order. The surface of $Sr_2RuO_4$ is known to exhibit an octahedral rotation, with low energy electron diffraction measurements placing the angle at $\theta = 8.5 \pm 2.5°$[26], close to the critical angle of $9°$ at which the $\overline{M}$-point VHS becomes higher-order. To establish the low energy model that describes the experimental electronic structure in the vicinity of the VHS at the $\overline{M}$ point, we perform a polynomial interpolation of the DFT-derived tight-binding models as a function of angle and a least squares fitting to ARPES data of the surface electronic structure[27], presented in Fig. 3a. We perform a grid-based search in a

parameter space that includes octahedral rotation, $\theta$, SOC $\lambda$, and a quasiparticle renormalisation factor, $Z$. The best fit to the ARPES bands in the vicinity of the $\overline{M}$ point is obtained for $\theta = 8.03°$, $\lambda = 0.17eV$, and $Z = 0.24$. We plot the obtained model over the ARPES data in Fig. 3b, and show this together with the extracted data points from the ARPES measurements used for the fitting in Fig. 3c. The model shows excellent agreement with the dispersions extracted from the ARPES data in the vicinity of the VHS at the $\overline{M}$ point (Fig. 3d), and the octahedral rotation angle $8.03°$ is in agreement with measurements of the surface structure[26].

## Comparison with QPI

The ARPES data reveal an important complication to the low-energy electronic structure of the surface of $Sr_2RuO_4$. Whilst they allowed us to locate the saddle point at around 18 meV below $E_F$, the bands in the vicinity of $E_F$ are found to be dominated by a spin–orbit gap in the electronic structure which can also give rise to VHS in the DoS[27,29]. We note that spin–orbit coupling has also been shown to play a crucial role in the bulk electronic structure of $Sr_2RuO_4$[24,31]. To gain further insight into the VHS mediated by this spin–orbit gap, we refine our experimental model of the VHS by comparing with QPI data obtained by STM at millikelvin temperatures and with sub-meV energy resolution. QPI provides access to the electronic states above and below $E_F$. Low-energy low-temperature differential conductance measurements of the surface of $Sr_2RuO_4$ were previously reported[20,32] and established the existence of a partial gap with four peaks in the vicinity of $E_F$, indicating the presence of four close-lying VHS. However, when we compare a theoretical continuum LDoS calculation of the tunnelling spectrum $g(V)$ from the tight-binding model obtained by fitting to the ARPES data, 3(b), to the STM measurements (shown in grey and black, respectively, in Fig. 4a), we find that although the model captures the partial gap, it only predicts two of the observed four peaks, and the position of the partial gap is several meV too low. The QPI data exhibits a small nematic term breaking the four-fold rotation symmetry of order $\sim 1 meV$[20,32]. Incorporating this nematicity via an anisotropy of the nearest-neighbour hopping into the tight-binding model obtained from the fit to the ARPES data, and adjusting the chemical potential by $\sim 3 meV$, we reproduce the experimental tunnelling conductance with sub-meV resolution. The small energy shift originates from the model fit to the ARPES data being only constrained to the occupied states, whereas QPI constrains the model in addition in the unoccupied states, as well as from technical aspects such as different measurement temperatures. To confirm the accuracy of this model, we perform continuum QPI calculations, modelling scattering of electrons from a point-like defect, and compare with experimental data[32] (see Supplementary Information for details). The calculation is in excellent agreement with the key features of the experimental data within a $\pm 5$ meV window. Fig. 4b shows a comparison between calculation and experiment of a representative constant energy cut, while Fig. 4c, d displays cuts as a function of energy.

We find that the formation of the gap-like structure close to $E_F$ can be understood completely as due to the spin–orbit gap found in ARPES. By calculating the atomic corrugation of the real space LDoS using our new model, we find that SOC, coupled with a finite nematicity, produces the experimentally observed checkerboard modulation (Fig. 4e, f)[20]. In previous works[20,32] both nematicity and checkerboard charge order had to be introduced to reproduce the experimental data, whereas here the experimental data is well described without the need to explicitly introduce a checkerboard term into the tight-binding model. It further shows that the $g$-factor $g^* \approx 3$ of the Zeeman shift of the lower VHS observed previously[20] can be attributed to the $d_{xy}$ VHS. Our tight-binding model unifies ARPES and STM measurements of the low-energy electronic structure of the surface of $Sr_2RuO_4$ with sub-meV energy resolution.

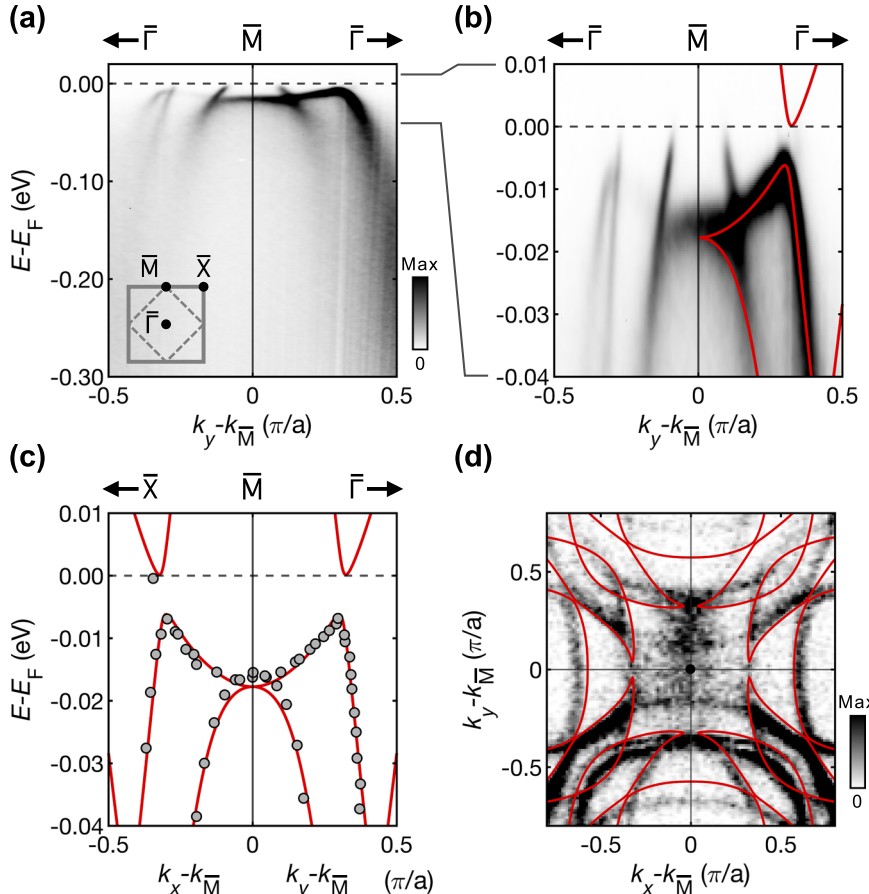

**Fig. 3 | Electronic structure from ARPES and fitting of the tight-binding model.** **a** Band dispersions of the bulk and surface electronic structure of Sr₂RuO₄ measured by ARPES along the $\bar{\Gamma} - \bar{M}$ direction of the surface BZ, reproduced from ref. 27. **b** Zoom-in of the measured electronic structure near $E_F$, with the fitted tight-binding model shown atop (red lines, $\theta = 8.03°$, $\lambda = 0.17$eV, and $Z = 0.24$). **c** The model is fitted to the bands extracted from the measurements shown in (**b**) and from additional measurements (ref. 27). **d** Corresponding Fermi surface measured by ARPES and calculated from our tight-binding model. The colour bars in (**a**) and (**d**) indicate the photoemission intensity in arbitrary units.

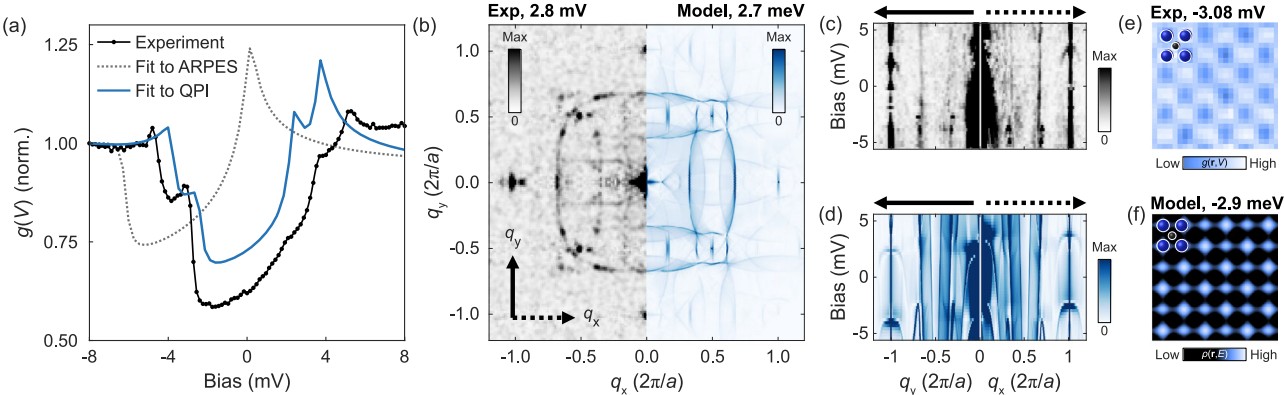

**Fig. 4 | Low-energy electronic structure from QPI.** **a** Comparison of experimental and calculated tunnelling spectrum of the clean surface, for the models fit to ARPES (grey, dotted) and to QPI (blue). **b** Left half: Quasi-particle interference image at $E = 2.8$mV, right half: calculations using model fitted to experimental data. Cuts through the atomic peaks along $q_y$ (left) and $q_x$ (right) from experiment (**c**) and calculations for the tight-binding model fitted to the QPI (**d**). Real-space image showing the checkerboard pattern at the Sr positions, from experiment (**e**) and the model (**f**). Blue and black spheres indicate the position of the Sr and Ru atoms, respectively.

## Order of the VHS

Our observation of a SOC-induced gap at $E_F$ raises the question what the type of the VHS at the gap edge is. Figure 5a, b presents the band structure, and corresponding DoS, of our final optimised model. This model is quantitatively accurate around the $\bar{M}$ point, now labelled by two subscripts due to the small nematic splitting rendering the $\bar{M}$ points nonequivalent along the Ru-Ru reciprocal lattice directions.

We proceed to characterise the VHS found within 20 meV of $E_F$ around the $\bar{M}$ points taking into account the nematicity. The nematicity changes the critical angle where the HOVHS with the divergent

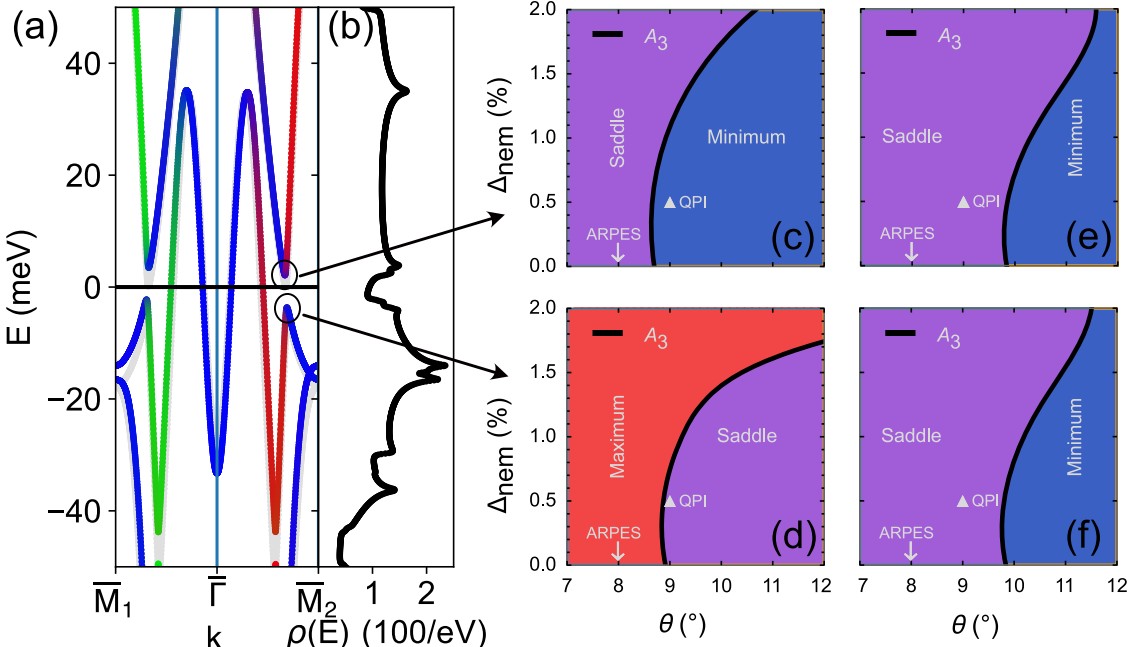

**Fig. 5 | Analysis of spin–orbit induced VHS in the surface layer of Sr$_2$RuO$_4$.**
**a** Band structure and (**b**) DoS around $E_F$ for the tight-binding model fitted to the QPI. Light grey lines in (**a**) show the band structure fitted to ARPES. **c, d** Phase diagrams of the VHS closest to the $E_F$ at the upper and lower edge of the SOC-induced gap, marked by circles in (**a**) as the octahedral rotation $\theta$ and the percentage nematicity $\Delta_{nem}$ are tuned. Solid lines indicate the line where the VHS becomes an $A_3$ singularity. **e, f** corresponding phase diagrams for the singularities at $\overline{M}_2$.

exponent $-1/4$ (and class still A$_3$) is located. The critical line as a function of the percentage change of the nematicity, for values experimentally achievable through uniaxial strain, is computed (Fig. 5c, d). This line separates the regions of maximum/minimum and saddle point VHS, therefore the system can be tuned to an $A_3$ HOVHS by applying strain as a tuning parameter. Notably, for the SOC-induced VHS at negative energies, from our QPI data the VHS is very close to the line where it becomes A$_3$, demonstrating that with modest and experimentally achievable amounts of uniaxial strain it can be tuned there. The $d_{xy}$ VHS at the $\overline{M}$ point exhibits a similar phase diagram with octahedral rotation and uniaxial strain, see Fig. 5e, f.

## Discussion
In previous studies, the occurrence of HOVHS has been discussed in terms of symmetry arguments (in the case of Sr$_3$Ru$_2$O$_7$) or from model electronic structures (for twisted bilayer graphene). Here, we introduce an approach that is in principle fully ab-initio, to model the electronic structure and apply an assumption-free method to determine the order of the HOVHS that is applicable, in principle, to any material. We demonstrate application of this method both to DFT-derived band structures and band structures that have been refined to match experiment, demonstrating the generality of this method.

This has enabled us to achieve two things as we explain below. The first is to provide direct evidence and predictions for the existence of a HOVHS in a quantum material. The second is to identify experimentally accessible tuning parameters that can be employed to control the order of the VHS (rotating the octahedra and uniaxial strain).

In particular, we have analysed two sets of complementary experimental data (STM and ARPES) of the benchmark material Sr$_2$RuO$_4$. We are able to classify and understand the nature of HOVHS close to the Fermi surface. The work also refines a long-standing question on the appearance of HOVHS in Sr$_2$RuO$_4$, as evidenced through ARPES measurements very early[33,34], before surface and bulk contributions were distinguished[23].

The combined theoretical and experimental characterisation provides key results: (i) the model for the low-energy electronic structure, consistent with STM and ARPES, captures the dispersion related to the VHS. While the comparison with STM requires a small energy shift, the amended model fitted to the QPI data still provides very good agreement also with ARPES. It provides an accurate description of the gap-like structure seen in tunnelling spectra that were previously interpreted in terms of inelastic tunnelling or correlation effects[25] or a hybridization gap[20,32]. This gap can be explained straightforwardly as the combined result of SOC and octahedral rotation, leading to a suppression of DoS by almost 50%. Apart from a renormalisation of the band structure by almost a factor of 4 compared to DFT calculations, the experimental band structure is largely consistent with the one obtained from DFT. Our low-energy model promises an understanding of the, yet unexplained, suppression of superconductivity in the surface layer of Sr$_2$RuO$_4$[20,35,36]. It will also provide a better basis to model the recently reported signatures of 'demons' measured in surface-sensitive electron loss spectroscopy measurements of Sr$_2$RuO$_4$[37], and to understand the experimental consequences of loop currents[38]. (ii) From the series of tight-binding models used to fit the experimental data, we can deduce design criteria for stabilising HOVHS in Sr$_2$RuO$_4$. In addition, in the SM we show that instead of strain, when a staggered chemical potential that makes two $d_{xy}$ bands nonequivalent is used, a four-fold symmetric $X_9$ singularity with a stronger divergent DoS exponent of $-1/2$ can be engineered (Fig. 1 of SM). Engineering of a HOVHS requires the tuning of at least one externally controlled parameter. The effect of these parameters is either to merge pieces of the Fermi surface at high symmetry points in the Brillouin zone or to make flat bands disperse and then leading to HOVHS. The important message of this study is that depending on the tuning parameter, different types of HOVHS can be realised.

To this end, the introduced set of analytical and numerical tools enables the determination of the order of the HOVHS, leading to a phase diagram of their order. The $d_{xy}$-VHS clearly changes it type as a function of rotation angle, with a critical angle of ~9.7°, surprisingly close to the angle realised in the surface reconstruction of Sr$_2$RuO$_4$. Tunability of the rotation angle can be achieved by biaxial strain, as demonstrated in MBE-grown thin films, where epitaxial strain and

substitution of Sr by Ba lift the octahedral rotation[39], or as suggested to result from electric fields through adsorption of alkali atoms[40]. In combination with doping or gating to control the chemical potential, thin films of strontium ruthenates become a promising testbed to establish the influence and importance of HOVHS for the physical properties.

As we have emphasized, HOVHS and the corresponding flattening of the single-particle band structure have an important role in the search of new quantum states of matter and unconventional thermodynamic and transport properties, due to the crucial role that interactions play in quantum materials with small bandwidth. Even weak interactions may lead to a plethora of quantum phases as a result of HOVHS[41–43]. Advances in experimental techniques make it possible to tune systems to nearly flat bands/HOVHS, as a result highly accurate experiments can probe the physics of HOVHS and flat bands. The theoretical and numerical tools we introduce here provide a methodology to study tuneable VHS in a range of existing quantum materials (e.g., ref. [44]) and in many more predicted to exhibit almost flat bands[45]. Due to the general applicability of the methods introduced here we believe that the impact is beyond the specialized area as the work provides a platform to engineer HOVHS and study their consequences in real materials. The next improvement of our approach to use DFT-derived tight-binding models is to account for many-body corrections that might be expected near a diverging DoS. This can be achieved if our method is combined with DMFT-derived band structures that incorporate self-energy effects to obtain better estimates e.g., of critical exponents. It is also worth noting that the same methods could be applied to other quasiparticle band structures, such as, e.g., phonons or polaritons.

## Methods

### Experimental data

All details regarding the experimental data reproduced in this manuscript can be found in their original publications ref. [27] (Fig. 3) and ref. [20] (Fig. 4).

### Density functional theory calculations

DFT calculations were performed using the augmented plane-wave code Quantum Espresso[46] using the PBE for the exchange-correlation functional. We used a monolayer of $Sr_2RuO_4$ with 15Å of vacuum on either side for each calculation with the planar oxygen atoms within the $RuO_6$ octahedra rotated by angles between 0 and 12° to study the role of octahedral rotation on the electronic structure. The energy cutoff for the wavefunction and charge density was chosen at 90 Ry and 720 Ry respectively and a $\mathbf{k}$-grid of $8 \times 8 \times 1$ was used for each calculation. It can be shown from relaxation of slabs as well as monolayers that such octahedral rotation is stabilised at the surface of $Sr_2RuO_4$ and can be controlled by adsorbates[40] or biaxial strain.

### Tight-binding models

These DFT band structures were then projected onto an orthonormal tight binding basis consisting of the Ru $d_{xz}$ $d_{yz}$ and $d_{xy}$ orbitals in a two-atom unit cell. This was achieved using a modified version of Wannier90[28] where we fix the phase of the Wannier functions against the first orbital to preserve the sign of the wave functions between neighbouring Ru atoms. This Wannierisation was performed on a $7 \times 7 \times 1$ $\mathbf{k}$-grid.

For fitting the ARPES band structure using the tight-binding models, we perform a polynomial interpolation on the hopping strengths of the thirteen individual models obtained for angles $\theta = 0°$ through 12° enabling fitting of $\theta$ as a continuous variable. We add a local spin–orbit term $H_{SOC} = \lambda \mathbf{L} \cdot \mathbf{S}$. Lastly, we allow for an overall band renormalisation and shift of the chemical potential. Fitting the model to the ARPES band structure therefore implies optimising the

following free parameters: the SOC strength $\lambda$, RuO octahedral rotation angle $\theta$, overall band renormalisation $Z$ and the chemical potential $\mu$.

For comparison with the QPI results, we introduce a nematicity of 0.5% across all orbitals, multiplying the nearest neighbour hoppings in one direction across all orbitals by 1.01 and dividing it in the orthogonal direction by 1.01.

### Continuum Local DoS calculations

For continuum local DoS (LDoS) calculations shown in Fig. 4, we use the tight-binding model obtained from the fit to the ARPES data modified to be consistent with the STM differential conductance. We use the localized wave functions obtained at an angle close to the one obtained from the fit. Because the tunnelling spectrum is significantly better described with a slightly larger octahedral rotation angle than the 8° obtained from the fit, we use as a basis the model and wave functions with $\theta = 9°$. The continuum LDoS calculations are performed using the method introduced in[47–49] and recently applied to ruthenates[32,50] using the St Andrews calcqpi code[22,50–52]. Calculations shown in Fig. 4 were done on a $\mathbf{k}$-space grid of $2048 \times 2048$ points distributed uniformly across the BZ of the two-atom unit cell. The real-space continuum LDoS is calculated over an area corresponding to $128 \times 128$ unit cells and with an energy broadening of 0.2 meV. We assume isotropic scattering with a scattering potential of $V = 1$ eV across all orbitals and spin directions.

### Numerical Integration of DoS

We calculate the DoS $\rho(E)$ from the unperturbed Green's function using $\rho(E) = -\text{Im} G_0(E)$ with an energy broadening of $\Gamma = 0.75$ meV, and using a $k$-space grid with $2088 \times 2088$ over the first BZ. The DoS is calculated on a regular grid with an energy spacing of 0.4 meV. To obtain the logarithmic derivative, we use a Savitzky–Golay algorithm with eleven points.

### Tuning to a HOVHS

We briefly describe the procedure to tune the VHS in the tight-binding models into HOVHS. We first choose a band and a critical point of interest to us. Let us denote them by $n$ and $\mathbf{k}^{(0)}$ respectively. If $\mathbf{k}^{(0)}$ is a critical point, then the gradient of the dispersion of band $n$ vanishes, i.e $\nabla \varepsilon_n(\mathbf{k}) = 0$ at $\mathbf{k} = \mathbf{k}^{(0)}$. If the Hessian determinant $\det[\partial_i \partial_j \varepsilon_n(\mathbf{k})]$ also vanishes at $\mathbf{k}^{(0)}$, we then have a HOVHS. Now the tight-binding Hamiltonian (and therefore the dispersion) also depend on the tuning parameters which we lump together as a vector $\mathbf{t}$. Therefore, the Hessian at any given $\mathbf{k}$ is a function of $\mathbf{t}$ as well. Thus, if we set the Hessian determinant evaluated at $\mathbf{k}^{(0)}$ to zero, we can solve for the critical value of $\mathbf{t}$ that gives us a HOVHS at $\mathbf{k}^{(0)}$. To diagnose the type of the HOVHS, we will need the series expansion of the dispersion to a higher degree than just quadratic order (which is sufficient for computing the Hessian).

Now this procedure is fairly straightforward in theory: we simply have to analytically diagonalise the band Hamiltonian $H(\mathbf{k})$ and obtain the $n^{\text{th}}$ band dispersion as a function of $\mathbf{k}$, denoted by $\varepsilon_n(\mathbf{k})$. We can then calculate the partial derivatives of $\varepsilon_n(\mathbf{k})$ or equivalently compute its series expansion at the chosen $\mathbf{k}^{(0)}$ to any desired degree. However, it is usually possible to implement this analytic procedure only for two band Hamiltonians. Multi-band, detailed Hamiltonians necessitate the use of an alternate, computationally implementable strategy for obtaining the series expansion (and the partial derivatives).

The crux of this alternate method lies in the realisation that we only need to compute the numerical values of the partial derivatives $\partial^{l+m} \varepsilon_n(\mathbf{k}) / \partial^l k_x \partial^m k_y$ evaluated at $\mathbf{k}^{(0)}$ (for $1 \leqslant l + m \leqslant N$), in order to construct the Taylor expansion to degree $N$. In the one dimensional case, the first derivative can be computed using the Hellman-Feynman

theorem which reads

$$\frac{d\varepsilon_n(k)}{dk}\bigg|_{k^{(0)}} = \left\langle n, k^{(0)} \left| \frac{dH(k)}{dk} \right|_{k^{(0)}} \right| n, k^{(0)} \right\rangle. \tag{1}$$

Note that this requires us to be able to (i) numerically obtain the eigenvector $|n, k^{(0)}\rangle$ and (ii) differentiate the Hamiltonian matrix $H(\mathbf{k})$, both of which can be readily done, even in the case of large multi-band Hamiltonians. Therefore, to compute the full Taylor expansion in the two and three dimensional cases, we have to work with an appropriate generalisation of the Hellman-Feynman theorem which can also handle complicated multi-band degeneracies that often prop up at high-symmetry points. A comprehensive treatment of such an extension can be found in ref. [15]. The final formula for the Taylor expansion up to second order reads

$$\varepsilon_n(\mathbf{k}^{(0)} + \mathbf{k}) \approx \left\langle n, \mathbf{k}^{(0)} \left| \frac{\partial H(\mathbf{k}^{(0)} + \lambda \mathbf{k})}{\partial \lambda} \right|_{\lambda = 0} \right| n, \mathbf{k}^{(0)} \right\rangle$$

$$+ \left\langle n, \mathbf{k}^{(0)} \left| \frac{\partial^2 H(\mathbf{k}^{(0)} + \lambda \mathbf{k})}{\partial \lambda^2} \right|_{\lambda = 0} \right| n, \mathbf{k}^{(0)} \right\rangle \tag{2}$$

$$+ \frac{1}{2} \sum_m \frac{\left| \left\langle n, \mathbf{k}^{(0)} \left| \frac{\partial H(\mathbf{k}^{(0)} + \lambda \mathbf{k})}{\partial \lambda} \right|_{\lambda = 0} \right| m, \mathbf{k}^{(0)} \right\rangle \right|^2}{\varepsilon_n - \varepsilon_m} + \mathcal{O}(k^3),$$

where $\mathbf{k} = (k_x, k_y)$ and the sum in Equation (2) is over other the eigenvectors $|m, \mathbf{k}^{(0)}\rangle$ that are not degenerate to $|n, \mathbf{k}^{(0)}\rangle$. Note that only $\mathbf{k}$ and $\lambda$ are symbolic variables while $\mathbf{k}^{(0)}$, $|n, \mathbf{k}^{(0)}\rangle$ and $|m, \mathbf{k}^{(0)}\rangle$ are numerical vectors. This is what makes the method viable for large and complicated models. By using the auxiliary parameter $\lambda$ in the formula above, we will directly obtain the series expansion as a multivariate polynomial in $k_x$ and $k_y$. The partial derivatives can be extracted from the Taylor coefficients.

Furthermore, we can also readily track the effect of tuning parameters on the low-energy expansion by using $H(\mathbf{k}^{(0)} + \lambda \mathbf{k}, \mathbf{t}^{(0)} + \lambda \delta\mathbf{t})$ in place of $H(\mathbf{k}^{(0)} + \lambda \mathbf{k})$ in the formula above, where $\mathbf{t}^{(0)}$ is the set of tuning parameters (their chosen, numerical values to be precise). The perturbations to $\mathbf{t}^{(0)}$ will be incorporated through the symbolic vector $\delta\mathbf{t} = (\delta t_1, \delta t_2, \cdots)$. In the present context, the tuning parameters of interest are the RuO octahedral rotation angle $\theta$ and the nematicity $\Delta_{\mathrm{nem}}$.

In this formulation, the numerical Taylor coefficients at the chosen $(\mathbf{k}^{(0)}, \theta^{(0)}, \Delta_{\mathrm{nem}}^{(0)})$ will be perturbed by powers of $\delta\theta$, $\delta\Delta_{\mathrm{nem}}$. We can then compute the Hessian determinant at $(\mathbf{k}^{(0)}, \theta^{(0)}, \Delta_{\mathrm{nem}}^{(0)})$. This will be a polynomial in $\delta\theta$ and $\delta\Delta_{\mathrm{nem}}$. We set it to zero to solve for $\delta\theta^*$ and $\delta\Delta_{\mathrm{nem}}^*$. The critical values of the tuning parameters that yield a HOVHS are then given by $\theta^{(c)} \approx \theta^{(0)} + \delta\theta^*$ and $\Delta_{\mathrm{nem}}^{(c)} \approx \Delta_{\mathrm{nem}}^{(0)} + \delta\Delta_{\mathrm{nem}}^*$. This process can be repeated a few times to obtain a more accurate value of the tuning parameters that lead to a HOVHS at $\mathbf{k}^{(0)}$.

The strategy outlined above works quite well at high-symmetry points like the $\overline{\mathrm{M}}$ wherein the location of the critical point remains fixed even as we change the tuning parameters. For the SOC gap VHS, we have to adopt a slightly different strategy. In this case, the location of the critical point $\mathbf{k}^{(0)}$ changes as we change $\theta$ and $\Delta_{\mathrm{nem}}$. For $\Delta_{\mathrm{nem}}$ in the range of 0% to 2%, we find that the Hessian determinant for the critical points in the SOC gap changes sign in the range of 7° to 12°. So for any given $\Delta_{\mathrm{nem}}$, we perform binary search to find the value of $\theta$ at which the Hessian determinant vanishes. At any intermediate $\theta$, we have to compute the new location of $\mathbf{k}^{(0)}$. To do this, we identify an interval $[\mathbf{k}^{(\mathrm{low})}, \mathbf{k}^{(\mathrm{up})}]$ that contains $\mathbf{k}^{(0)}$ and perform binary search along the line connecting the endpoints of this interval to locate $\mathbf{k}^{(0)}$ (the point where the directional derivative vanishes in this interval).

## Data availability

The data underpinning Figs. 3 and 4 have been published with refs. 27 and [20].

## Code availability

The code and tight binding model are available at the associated GitHub repository[53].

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

## Acknowledgements

We thank Andy Mackenzie and Andreas Rost for useful discussions. We are grateful to the UK Engineering and Physical Sciences Research Council for funding via Grant Nos EP/T034351/1, EP/T02108X/1, EP/R031924/1 and EP/S005005/1. LCR and PW further acknowledge support from the Leverhulme Trust through RPG-2022-315. C.A.M. was supported by the Federal Commission for Scholarships for Foreign Students for the Swiss Government Excellence Scholarship (ESKAS No. 2023.0017) for the academic year 2023-24. This work used computational resources of the Cirrus UK National Tier-2 HPC Service at EPCC (http://www.cirrus.ac.uk) funded by the University of Edinburgh and EPSRC (EP/P020267/1) and of the High-Performance Computing cluster Kennedy of the University of St Andrews.

## Author contributions

A.C. and J.J.B. set up the theoretical analysis. L.C.R. performed DFT calculations and projected the tight-binding models, C.A.M. analyzed QPI data and calculations, and P.W. performed the calculation of the DOS and its logarithmic derivative. E.A.M and P.D.C.K. provided the ARPES data and contributed to their analysis. A.C., L. R., P.W. and J.J.B. wrote the manuscript with input from all authors. All authors discussed the results. J.J.B. conceived and initiated the project.

## Competing interests

The authors declare no competing interests.
