## [Peer Review File · Nature Communications]

On the engineering of higher order Van Hove singularities in two dimensions.

Corresponding Author: Professor Joseph Betouras

Version 0:

Reviewer comments:

Reviewer #1

(Remarks to the Author)

Overall, I found the paper to be well-written with minor imperfections in figures. The authors applied a recently developed method, previously published by the same group, to the material Sr₂RuO₄. The novelty of the manuscript lies in the new approach of describing the established band structure using a combination of Density Functional Theory (DFT) and tight-binding calculations. While the methods provide a good agreement and the tight-binding approach uses only a few parameters for fitting the experimental structure, it does not provide a prediction of the band structure behavior. Additionally, including the consideration of other 2-dimensional materials would be helpful for a general reader to better understand how to engineer higher order Van Hove singularities.

The paper's main strength lies in its application of the new theoretical method to a real material. However, it is worth noting that ARPES data and band structure calculations for Sr₂RuO₄ have been previously presented, reducing the novelty of the current work. While the methodology is sound and meets the expected standards in the field, it would be beneficial for the authors to provide more details in the methods section to allow for reproducibility.

I would like to ask a question that remains unclear to me: how do the authors explain the small energy shift of the band structure in the fit applied to ARPES and STM data?

In summary, this paper is a strong contribution to a specialized journal. It would be valuable to see the expanded application of the model to other materials. Some improvements in the figures and additional explanations in the methods section regarding reproducibility would further strengthen the quality of the manuscript.

Reviewer #2

(Remarks to the Author)

The manuscript reports a theoretical analysis of higher-order Van Hove singularities (HOVHs) in a strontium ruthenate Sr₂RuO₄. The authors present an accurate theoretical modeling of ARPES and STM results, which reveals a good quantitative agreement. The result is actually surprising considering relatively strong electron correlation. The logic is clear and the method seems sound. Nevertheless, the presentation can be clearer. Also, a significant advance from previous works, particularly Phys. Rev. Lett. 123, 207202 (2019), is not evident, aside from the good quantitative agreement. The manuscript may finally be published in Nature Communications, but I suggest a more specialized journal at present after the comments below are answered.

1. Most importantly, the authors should describe the novelty of the work. Analyses of HOVHs have already been published in several papers. Probably the most related work to this manuscript is Phys. Rev. Lett. 123, 207202 (2019), which considered Sr₃Ru₂O₇. I understand that the authors figured out a very good agreement with experimental observations, with a unit cell rotation angle being a control parameter. The method, unfortunately, appears rather similar. Any comments on differences and updates would improve the novelty of the present work.

2. The authors repeatedly mentioned "our model" in the manuscript, though I could not find an explicit one. A continuum model near HOVHs should be useful. It reflects the symmetry at the HOVHs and describes the power law of the divergence of the density of states.

3. Perhaps, a consideration of the spin-orbit coupling is one new thing of this work. However, without an explicit model, the discussion seems ambiguous. I suggest that the authors present the details of their analysis.

4. As the spin-orbit coupling splits the original sharp HOVHs, the rapid increase of the density of state near the Fermi level is not evident. In Fig. 5(b), we can find higher peaks at about -10 meV. Any comment about its consequences would be useful.

5. In Fig. 2(c), the authors illustrate the type of a VHS. Its meaning is obscure. The boundary between log and power-law divergences is a crossover, I presume. I wonder how they determine the boundary. As such, I could not expect any sharp difference between the two regions.

6. The authors estimate the quasiparticle renormalization factor $Z=0.24$ from the ARPES data. It reveals a moderate electron correlation effect, which is not mentioned in the manuscript. They should comment on the correlation effect to justify the analysis.

Reviewer #3

(Remarks to the Author)

The authors investigate the high-order van Hove singularities (HOVHs) in the surface layer of Sr₂RuO₄ through DFT calculations, theoretical analysis, and fitting a tight-binding model to ARPES data. They find that HOVHs can be achieved at specific octahedral rotational angles in the DFT calculations. Additionally, they fit the ARPES dispersion to obtain a realistic tight-binding model, which can reproduce QPI data consistent with STM measurements. Finally, the authors demonstrate that HOVHs can be realized in this realistic model with SOC by varying the nematicity and rotational angle. The calculations are solid and conclusions are clear. However, the significance of this paper and its suitability for publication in Nature Communications is not entirely clear. The potential impact of HOVHs on the physical properties of Sr₂RuO₄ is not clearly articulated. Furthermore, the introduction mentions that HOVHs have been observed in various systems, such as twisted bilayer graphene and kagome metals, which raises questions about the novelty and broader significance of the findings. While the work is interesting and valuable, it may not meet the level of significant impact expected for publication in Nature Communications. The manuscript should address the following points for further clarification:

1. The band structures in Fig.1 are obtained from DFT calculations without considering SOC. It would be valuable for the authors to investigate the impact of SOC on the band evolution and HOVHs. The authors also need to provide a physical understanding for the band evolution with varying octahedral angles around the M point.

2. In Fig.2, the authors present the exponent of the DOS as a function of octahedral angles. To further directly illustrate the emergence of HOVHs, it would be beneficial to include the evolution of parameters of quadratic and quartic terms near the M point with varying rotation angles. The quadratic term along the M-Gamma direction is expected to vanish when the HOVHs emerge.

3. Figures 3 and 4 compare theoretical results from the fitted TB model with experimental data from ARPES and STM. It would be helpful for the authors to clarify whether these comparisons are related to the formation of HOVHs or if they are used to illustrate certain features of HOVHs.

4. The authors may need add some discussions of the physical implications of HOVHs, either specific to Sr₂RuO₄ or in a more general context. This will help to provide a clearer understanding of the potential impact of HOVHs.

Version 1:

Reviewer comments:

Reviewer #1

(Remarks to the Author)

This manuscript presents an exciting new approach to engineering high-order van Hove singularities (VHS) for enhanced material properties. The work represents a significant step forward in the development of this field and has the potential to lead to the creation of novel materials with tailored properties.

The manuscript is well-written and easy to follow. I am confident that this work will be of significant interest to the research community. Overall, I recommend this manuscript for publication with minor revisions.

Specific suggestions for improvement:

Provide a more detailed explanation of how the authors' approach differs from previous studies on engineering high-order VHS.

Include a more detailed description of the code used to perform the calculations.

Clarify the experimental setup.

Discuss the limitations of the current approach and future directions for research.

I am confident that the authors can address these suggestions and produce a manuscript that is of use to a wide audience.

Reviewer #2

(Remarks to the Author)

The authors revised the manuscript, reflecting the questions and comments from the reviewers. I found the improvements in the manuscript, but unfortunately, I could not see a significance that urged me to recommend publication in Nature Communications while acknowledging the detailed analysis for Sr₂RuO₄ concerning HOVHs.

They showed a systematic study of a tunable two-dimensional material. The method is, in principle, applicable to any two-dimensional material. It is correct; on the other hand, various interesting physics, including Pines' demon, involve electron interaction effects. A study of a single-particle band structure is undoubtedly important, but it is not apparent how its fine structure alters the result.

Speaking of the interaction effects, their result revealed an accurate agreement with ARPES and QPI data. As I asked in the previous report, I am not sure if the single-particle picture well describes the physics with the quasiparticle renormalization factor $Z=0.24$, corresponding to a rather strong correlation. Particularly, QPI involves scattering problems, which are also affected by impurities.

Any model for HOVHs must include one control parameter. As they added in SM Section 1, the parameter alpha should vanish for the DOS to obey a power law. This is like a critical point in the parameter space. However, in Fig. 2c they showed that the power-law behavior appears in certain ranges of the angle. If this is correct, a critical behavior sustains for a finite range of the angle, which is surprising. Nevertheless, I could not see a clear correspondence to Fig. 2d.

Reviewer #3

(Remarks to the Author)

I would like to express my appreciation to the authors for their great efforts in addressing the concerns I raised. In the revised manuscript, I am pleased to see that some of my concerns have been addressed. However, upon further review of the paper, I have observed that the connection between A and B, C is not as robust as expected. While theoretical DFT calculations are presented in the former, the results from tight-binding model fitting to experiments are presented in the latter, which may impact the coherence of the study. Although the technique developed in this paper shows promise for exploring HOVHs, I have reservations regarding the significance of the physics discussed, particularly considering previous theoretical studies on HOVHs in various systems. Therefore, I still have reservations about the suitability of the paper for publication in Nature Communications. A specialized journal may be a more appropriate venue for this work.

Version 2:

Reviewer comments:

Reviewer #2

(Remarks to the Author)

The authors responded to the previous questions and comments and updated the manuscript. The current manuscript describes the role of a higher-order Van Hove singularity in Sr₂RuO₄. Although most analysis is limited to noninteracting physics, it might advocate for higher-order Van Hove singularities to broad communities. I think that this work will be published in Nature Communications.

Reviewer #1 (Remarks to the Author):

We would like to thank Reviewer #1 for their time to review the manuscript, the positive remarks and the very constructive comments which we have addressed below.

While the methods provide a good agreement and the tight-binding approach uses only a few parameters for fitting the experimental structure, it does not provide a prediction of the band structure behavior. Additionally, including the consideration of other 2-dimensional materials would be helpful for a general reader to better understand how to engineer higher order Van Hove singularities.

The presented work describes a path to engineer higher order Van Hove singularities (HOVHS). The strategy followed was to describe in detail in a step-by-step fashion, the construction of the Hamiltonian (using a Wannierized model with input from ARPES and quasiparticle interference data which were then reproduced with high precision) and then to study their effect in producing a HOVHS. The final piece of the study, to use another tuning parameter such as nematicity (which could model the effect of strain on the material) and produce a “phase diagram” of the HOVHS is a key to engineer HOVHS in the materials in general. Always, engineering HOVHS requires the tuning of externally controlled parameters. The effect of these parameters is either to merge pieces of the Fermi surface at high symmetry points in the Brillouin zone or to make flat bands disperse and then leading to HOVHS. This is something we now clarify in the text, in the discussion section. We note that our approach is fully ab-initio, based on DFT-derived tight-binding models, and can hence be straightforwardly applied to any material, for the numerical calculation of the order of the VHS in any dimensionality. It is directly relevant and applicable also to, e.g., twisted bilayer graphene or kagome’ materials, however including them is beyond the scope of the present manuscript, because a proper discussion would need to include a comprehensive study of tuning parameters and robustness, as done here using strain and octahedral rotations.

The paper's main strength lies in its application of the new theoretical method to a real material. However, it is worth noting that ARPES data and band structure calculations for Sr_2RuO_4 have been previously presented, reducing the novelty of the current work. *The choice to study Sr_2RuO_4 is precisely because the electronic structure is already well understood. This enables us to make conclusive statements about the order of the VHS and how to engineer them to higher orders, which is where the novelty of this manuscript lies. This is particularly important as our analysis shows that, impressively, small changes in the electronic structure can result in dramatic changes to the properties of the VHS.*

While the methodology is sound and meets the expected standards in the field, it would be beneficial for the authors to provide more details in the methods section to allow for reproducibility.

We note that not only is the theoretical method to extract the order of the VHS from a realistic tight-binding model new, but also the numerical technique to use the logarithmic derivative to determine the order of a HOVHS. The latter is completely model independent and can be applied to either experimental or theoretical densities of states and serve as a

diagnostic tool to detect HOVHSs. We have now included the code to determine the order from the series expansion with detailed explanations in a GitHub repository to allow reproducibility. We have also included a brief discussion of the method to estimate the crossover from log to power law in the vicinity of a HOVHS.

I would like to ask a question that remains unclear to me: how do the authors explain the small energy shift of the band structure in the fit applied to ARPES and STM data?

We note that it is in our view impressive to see that a model can be fitted with such a good agreement simultaneously to QPI and ARPES data. We are not aware of other examples in the literature where this has been achieved to this level, i.e. with an energy resolution of a few millielectronvolts. The origin of the energy shift required here is most likely that the tight-binding model fit to the ARPES data is only constrained in the occupied states, whereas STM probes both occupied and unoccupied states, which means, in particular, that the full spin-orbit gap needs to be captured. We also note that ARPES data was taken at 10K, with an energy resolution of a few millielectronvolts, whereas QPI data has been acquired at 100mK, with an energy resolution of approximately 100 microelectronvolts. We have added a sentence in the manuscript that points out these two aspects.

In summary, this paper is a strong contribution to a specialized journal. It would be valuable to see the expanded application of the model to other materials. Some improvements in the figures and additional explanations in the methods section regarding reproducibility would further strengthen the quality of the manuscript.

Because of the general applicability of the methods introduced here we believe that the impact is beyond the specialized area. In fact, the same methods could be applied also to other quasiparticle band-structures, such as, e.g., phonons or polaritons.

Reviewer #2 (Remarks to the Author):

We would like to thank Reviewer #2 for their time to review the manuscript, the positive comments and the very constructive questions which we have addressed in detail below.

The result is actually surprising considering relatively strong electron correlation. The logic is clear and the method seems sound. Nevertheless, the presentation can be clearer. Also, a significant advance from previous works, particularly Phys. Rev. Lett. 123, 207202 (2019), is not evident, aside from the good quantitative agreement. The manuscript may finally be published in Nature Communications, but I suggest a more specialized journal at present after the comments below are answered.

1. Most importantly, the authors should describe the novelty of the work. Analyses of HOVHs have already been published in several papers. Probably the most related work to this manuscript is Phys. Rev. Lett. 123, 207202 (2019), which considered Sr₃Ru₂O₇. I understand that the authors figured out a very good agreement with experimental observations, with a unit cell rotation angle being a control parameter. The method,

unfortunately, appears rather similar. Any comments on differences and updates would improve the novelty of the present work.

We would like to clarify this point which brings an impression that it is not correct. In the publication Phys. Rev. Lett. 123, 207202 (2019) one of us explained rather qualitatively the unusual experimental results by introducing the concept of multicritical Fermi surface topological transition. The quantitative results there were (i) the confirmation of the existence of this topological transition through DFT calculations without any further analysis and in a very broad sense (ii) that the energy dispersion taken, led to the correct specific heat dependence on the applied magnetic field. In the present work, we take for the first time a system of wide interest and analyse in detail the band structure as a function of a control parameter, characterising the HOVHS with new precise tools. We expect that this methodology that we introduce will be widely applied to all two-dimensional quantum materials. One control parameter is needed in every case, here is the octahedral rotation angle (that can be tuned through strain), in the case of Phys. Rev. Lett. 123, 207202 (2019) it was the magnetic field. We have added a small paragraph to clarify this point in the manuscript.

We further note that in view of the plethora of unconventional states that are claimed to exist in Sr_2RuO_4 , most recently a 'Pines demon' and loop currents, a good understanding of the low energy electronic structure will be crucial for realistic modelling of these states and to separate truth from fiction.

2. The authors repeatedly mentioned "our model" in the manuscript, though I could not find an explicit one. A continuum model near HOVHs should be useful. It reflects the symmetry at the HOVHs and describes the power law of the divergence of the density of states.

'Our model' refers to the tight-binding model that has been fitted to the experimentally determined band structure. We have clarified this now in the text. The parameters of that model are mentioned in the main text. The original Wannierized model is now included in the GitHub repository along with the code used to produce the results.

3. Perhaps, a consideration of the spin-orbit coupling is one new thing of this work. However, without an explicit model, the discussion seems ambiguous. I suggest that the authors present the details of their analysis.

We thank the referee for the comment; indeed, this is the first time that our method has been applied to understand spin-orbit induced HOVHs. We have now included the model in the located in the GitHub repository along with the detailed analysis of the spin-orbit gap HOVHs featured in the phase diagram in the Mathematica notebook '2_HOVHS.nb'.

4. As the spin-orbit coupling splits the original sharp HOVHs, the rapid increase of the density of state near the Fermi level is not evident. In Fig. 5(b), we can find higher peaks at about -10 meV. Any comment about its consequences would be useful.

The increase at -10meV is due to the dxy Van Hove singularity. We note that this VHS only comes close to the Fermi energy with uniaxial strain, where the SOC-induced VHS is much closer to the Fermi energy. We have added a comment on its relevance in the main text, and have added an analysis of its properties in fig. 5d, e.

5. In Fig. 2(c), the authors illustrate the type of a VHS. Its meaning is obscure. The boundary between log and power-law divergences is a crossover, I presume. I wonder how they determine the boundary. As such, I could not expect any sharp difference between the two regions.

We have now included a section in the Supplemental Material discussing the crossover from log to power law DOS in the vicinity of a HOVHS

6. The authors estimate the quasiparticle renormalization factor $Z=0.24$ from the ARPES data. It reveals a moderate electron correlation effect, which is not mentioned in the manuscript. They should comment on the correlation effect to justify the analysis.

We thank the reviewer for their comment; indeed this correlation factor is in very good agreement with previous literature comparing DFT band-structure calculations and ARPES data for the bulk bands of Sr_2RuO_4 [Ref 21. Tamai et al. PRX, 9, 021048 (2019)]. We have included a paragraph to this end in the main text.

Reviewer #2 (Remarks on code availability):

I did not find supplementary materials associated with the manuscript.

We have now included the code and the analysis in a GitHub repository.

Reviewer #3 (Remarks to the Author):

We would also like to thank Reviewer #3 for their time to review the manuscript, the positive comments and the constructive criticisms which we have addressed in detail below.

The calculations are solid and conclusions are clear. However, the significance of this paper and its suitability for publication in Nature Communications is not entirely clear. The potential impact of HOVHS on the physical properties of Sr_2RuO_4 is not clearly articulated. Furthermore, the introduction mentions that HOVHS have been observed in various systems, such as twisted bilayer graphene and kagome metals, which raises

questions about the novelty and broader significance of the findings. While the work is interesting and valuable, it may not meet the level of significant impact expected for publication in Nature Communications. The manuscript should address the following points for further clarification:

1. The band structures in Fig.1 are obtained from DFT calculations without considering SOC. It would be valuable for the authors to investigate the impact of SOC on the band evolution and HOVHs. The authors also need to provide a physical understanding for the band evolution with varying octahedral angles around the M point.

The band structure in Fig. 1 does include the influence of SOC, (as seen by a splitting of the bands above the Fermi level at the Gamma point) SOC is included here as a local SOC term at the tight-binding level and is in agreement with the previous literature e.g ref. 24 (Tamai et al. PRX, 9, 021048 (2019)) and ref. 27 (PRL130, 096401). With regards to the evolution of the VHS at the M point. While our model is derived from a DFT calculation, similar conclusions are reached when basing the model on orbital overlaps, including the octahedral rotations as done in ref. 26. The physical understanding here is that the rotation induces a hybridisation between the previously orthogonal dxz/dyz and dxy orbitals, which pushes the VHS down in energy. For illustrative purposes we present the analysis of the order of the VHS without SOC in Fig. 2, and the corresponding analysis with SOC in Fig. 5.

2. In Fig.2, the authors present the exponent of the DOS as a function of octahedral angles. To further directly illustrate the emergence of HOVHs, it would be beneficial to include the evolution of parameters of quadratic and quartic terms near the M point with varying rotation angles. The quadratic term along the M-Gamma direction is expected to vanish when the HOVHs emerge.

We have now included the evolution of the series expansion for both the M point singularity and the SOC gap singularities in the Mathematica notebook '2_HOVHS.nb' located in the GitHub repository.

3. Figures 3 and 4 compare theoretical results from the fitted TB model with experimental data from ARPES and STM. It would be helpful for the authors to clarify whether these comparisons are related to the formation of HOVHs or if they are used to illustrate certain features of HOVHs.

The purpose of figures 3 and 4 is to establish the surface electronic structure of Sr_2RuO_4 , to be able to make statements about the VHS in the surface layer. Our results provide an extremely well constrained model of the low energy electronic structure, therefore allowing for a detailed determination of the type of HOVHs that are stabilized in the surface layer. This is relevant, because it has previously been argued that the octahedral rotation and doubling of the unit cell facilitates the formation of HOVHs – the surface layer of Sr_2RuO_4 provides an ideal test system where this is realized. We now clarify this in the section explaining the structure of the manuscript.

4. The authors may need add some discussions of the physical implications of HOVHSs, either specific to Sr₂RuO₄ or in a more general context. This will help to provide a clearer understanding of the potential impact of HOVHSs.

HOVHS and the corresponding flattening of the single-particle band structure has an extremely important role in the search of new quantum states of matter due to the crucial role that interactions play in such situation, with the Fermi velocity tending to zero. Recent advances in theory and experiment made it possible to tune systems to nearly flat bands/HOVHS, as a result highly accurate experiments can probe the physics of HOVHS and flat bands. The systems that have been widely studied include graphene multilayers, moire' materials, kagome' metals and ruthenates.

The present work offers the first demonstration of a thorough analysis using new theoretical tools of the emergence and tuning of HOVHS in a quantum material, with potential application to any system with a band-structure. This has been emphasized now in the text.

Reviewer #1:

We would like to thank Reviewer #1 once more, for their time to review the manuscript again, the very positive remarks and the very constructive comments which we have addressed below.

1. Provide a more detailed explanation of how the authors' approach differs from previous studies on engineering high-order VHS.

In previous studies, the occurrence of HOVHS has been discussed in terms of symmetry arguments (in the case of Sr₃Ru₂O₇) or from model electronic structures (for twisted bilayer graphene). The key difference here is that we use a fully ab-initio approach to model the electronic structure and apply an assumption free method to determine the order of the HOVHS that is applicable, in principle, to any material.

This has enabled us to do two novel things in this manuscript: (a) to provide direct evidence and predictions for the existence of a HOVHS in a quantum material. (b) to identify experimentally accessible tuning parameters that can be employed to control the order of the VHS (rotating the octahedra and uniaxial strain), providing direct predictions.

This goes far beyond just identifying HOVHS in models as it provides an experimental platform to engineer HOVHS and study their consequences in real materials. In future, this will enable design of band structure with arbitrary types of VHS, and with potentially exciting consequences for the properties of these materials.

2. Include a more detailed description of the code used to perform the calculations.

We have recently uploaded the source code, along with instructions on how to use the code, in the GitHub directory https://github.com/anirudhc-git/VHS_Sr2RuO4. This is included in the code availability section. Each Mathematica notebook in the repository also contains detailed step by step instructions on usage and execution.

3. Clarify the experimental setup.

The experimental setups are not described in detail here, because the data presented in the manuscript has been published previously and only been analysed here for the purpose to search for HOVHSs. The information regarding the experimental details can be found in the original references. To make this clear to the readers we have now added a new section in the Methods (A. Experimental data).

All details regarding the experimental data reproduced in this manuscript can be found in the original publications Ref. [55] (Figure 3) and Ref. [20] (Figure 4).

4. Discuss the limitations of the current approach and future directions for research.

Our approach to use DFT-derived tight-binding models currently does not account for many-body corrections that might be expected near a diverging density of states. Hence, the next step would be to combine our method with DMFT-derived band structures that incorporate self-energy effects to obtain better estimates of critical exponents obtained from these models as would be relevant, for example, in materials such as Sr₃Ru₂O₇. Another direction that would be highly promising is to establish how the order of the Van Hove singularity affects electronic instabilities in a given material. The structural sensitivity which we find here strongly suggests that materials with structurally tuneable VHs will exhibit intricate electron-lattice couplings which likely will dominate the materials properties if the VHs approaches the Fermi energy. Finally, technically the method as it has been developed demands Wannierisation of the DFT results, therefore an extension to cases where Wannierisation may not be possible is desired. We have included these considerations in the discussion section of the manuscript.

Reviewer #2:

We would like to thank Reviewer #2 for their time to review the manuscript, the positive comments and the constructive questions which we have addressed in detail below.

1. I found the improvements in the manuscript, but unfortunately, I could not see a significance that urged me to recommend publication in Nature Communications while acknowledging the detailed analysis for Sr₂RuO₄ concerning HOVHs.

Key aspects of our work, which we believe merits publication in Nature Communications, are: (a) an assumption-free method to determine the order of VHS in arbitrary band structures, enabling a designer approach to create HOVHS and tune the materials properties controlled through them, (b) demonstration of a strong coupling between structural degrees of freedom and the properties of a HOVHS which will have important consequences for the ground state of the material – a link that has to the best of our knowledge not been established previously, but will result in new design principles. In establishing these, we chose the material Sr₂RuO₄ which has been attracting much interest, as a fundamental system to be understood, for decades.

2. They showed a systematic study of a tuneable two-dimensional material. The method is, in principle, applicable to any two-dimensional material. It is correct; on the other hand, various interesting physics, including Pines' demon, involve electron interaction effects. A study of a single-particle band structure is undoubtedly important, but it is not apparent how its fine structure alters the result.

The Referee is correct that this method can in principle be applied to any two-dimensional material, but it is not guaranteed that a generic two-dimensional material would host a HOVHS, and if it does whether that HOVHS will be close to the Fermi level,

which is a requirement if one wants to study and tune novel ground states induced by HOVHS's. With regards to interactions - our model is focused on the electronic structure within 20 meV of the Fermi level, in a Fermi liquid picture, this is very low energy, and hence the quasiparticle lifetimes are still significant, reaching infinity at the Fermi level, hence the single particle picture is still valid. Whilst we fully agree that interactions may alter the stabilisation of new interesting physics, the underlying ingredients still originate from the single particle description, and particularly the divergence of the single-particle VHS.

We agree though that the next steps would be to incorporate self-energy effects, obtained, e.g., from DMFT calculations, to model how (a) the HOVHS modify the electronic correlations and (b) incorporate the change in the density of states due to the self-energy corrections. The latter will have important implications, e.g., for thermodynamic properties. We have now highlighted this point in the discussion section of the manuscript.

3. Speaking of the interaction effects, their result revealed an accurate agreement with ARPES and QPI data. As I asked in the previous report, I am not sure if the single-particle picture well describes the physics with the quasiparticle renormalization factor $Z=0.24$, corresponding to a rather strong correlation. Particularly, QPI involves scattering problems, which are also affected by impurities.

There are several points we would like to highlight here – Firstly, this “Z” value is a qualitative indicator for correlations in so far as it discusses the agreement/disagreement with the bandwidth predicted from DFT. In that regard, this value is in excellent agreement with previous reports (e.g. Tamai et al.) and effective mass renormalisations inferred from de Haas van Alphen measurements. However, there is always some error with DFT and, as a matter of principle, this isn't necessarily indicative of strong correlations that break Fermi liquid theory. Additionally, as we discussed in the previous comment, here we focus on low energies close to the Fermi level, where the single particle description is valid. The referee is correct that there are regimes where the quasi-particle picture breaks down, for Sr₂RuO₄ this is the case once going further away from the Fermi energy. Otherwise, the material serves as a paradigm of a good Fermi liquid.

With regards to QPI and interactions, a key indicator of correlations is the linewidth that smears out the single particle band description. In that case QPI scattering vectors would too be smeared out and no sharp peaks should be observed in experimental measurements of the Fourier transform. As we show in Fig 4(b), this is not the case. Our quantitative agreement between the experimental data and the QPI calculations therefore a posteriori justify the approach, particularly at energies close to the Fermi level. We note, however, that the QPI becomes rapidly washed out with increasing

energy, suggesting that correlation effects become important with an associated breakdown of the quasi-particle description.

4. Any model for HOVHs must include one control parameter. As they added in SM Section 1, the parameter α should vanish for the DOS to obey a power law. This is like a critical point in the parameter space. However, in Fig. 2c they showed that the power-law behavior appears in certain ranges of the angle. If this is correct, a critical behavior sustains for a finite range of the angle, which is surprising. Nevertheless, I could not see a clear correspondence to Fig. 2d.

*As we show in SM Section 1, the **asymptotic behaviour to leading order is dominated by power law** for a range of energies close to the critical point. Power law DOS proper is indeed obtained only for the finely tuned case. A similar behaviour is seen when a HOVHS hosting material is doped. As two of us showed in Phys. Rev. B 105, 075144 (2022) for a range of impurity concentrations and scattering strengths, the leading order power law DOS is recovered for energies in the vicinity of the critical energy. Regarding the Fig. 2d, the asymmetric results above and below the Fermi level is due to the fact that at values of θ away from the critical one, there is deformation of the nearby bands with some energy gaps that get bigger (for example in the band-structure of Fig. 1 for θ around 12 degrees). This is reflected in Fig. 2d where white areas appear below the Fermi level and for values of θ around 11-12 degrees.*

Reviewer #3:

We would also like to thank Reviewer #3 for their time to review the manuscript, the positive comments. We believe the concerns they have raised is due to a misunderstanding of the approach we follow as we have indicated below.

I would like to express my appreciation to the authors for their great efforts in addressing the concerns I raised. In the revised manuscript, I am pleased to see that some of my concerns have been addressed. However, upon further review of the paper, I have observed that the connection between A and B, C is not as robust as expected. While theoretical DFT calculations are presented in the former, the results from tight-binding model fitting to experiments are presented in the latter, which may impact the coherence of the study. Although the technique developed in this paper shows promise for exploring HOVHS, I have reservations regarding the significance of the physics discussed, particularly considering previous theoretical studies on HOVHS in various systems. Therefore, I still have reservations about the suitability of the paper for publication in Nature Communications. A specialized journal may be a more appropriate venue for this work.

We believe there is a misunderstanding which led the referee to have reservations. It is true that Results A (Fig 1 and Fig 2) discuss DFT calculations, whereas the final results

and main conclusions are discussed from the developed tight binding model fitted to experiment. We note though that the tight-binding model is obtained from the DFT calculations and then fitted to the experiment. This is necessary because of the limitations of DFT, i.e. it does not properly capture the electronic structure in strongly correlated electron materials. The strength of our method is that it works with arbitrary band structure, whether they are obtained from a DFT calculation, a tight-binding model or a simple model. Here, using a band structure that is optimized to describe what is observed in experiment is crucial to verify the existence of a HOVHS. By optimising the DFT-derived model to match the experimental data we can be certain that our model does describe a real quantum material and thus allows for more robust conclusions. This would not have been possible with only a DFT-derived band structure.

Again, we wish to highlight that HOVHS are a topic of great current interest, as evidenced by the recent review by one of the authors to appear in Annual Reviews [arXiv:2405.20226]. Whilst so far theoretical work has focused on identifying HOVHS and their effects in toy models and from symmetry arguments, we here provide a general method to identify them in band structures and provide direct evidence for the existence of HOVHS in a strongly correlated electron material with specific guidance of how to tune the order of the VHS. This represents a significant step forward in the development of this field and has the potential to lead to the realisation of novel materials and ground states with tailored interactions.